# The contribution of fires to $PM_{2.5}$ and population exposure in Asia Pacific

Hua Lu[1,6], Min Xie[2], Nan Wang[3], Bojun Liu[4], Jinyue Jiang[5], Bingliang Zhuang[6], Ying Zhang[7], Meixuan Wu[7], Jianfeng Yang[8], Kunqin Lv[9], Danyang Ma[2]

Chongqing Institute of Meteorological Sciences, Chongqing 401147, China
School of Environment, Nanjing Normal University, Nanjing 210023, China
College of Carbon Neutrality Future Technology, Sichuan University, Chengdu 610065, China
Chongqing Meteorological Observatory, Chongqing 401147, China
The First Affiliated Hospital of Chongqing Medical University, Chongqing 400010, China
School of Atmospheric Sciences, Nanjing University, Nanjing 210023, China
School of Atmospheric Sciences, Chengdu University of Information Technology, Chengdu, 610225, China
The People's Hospital of Kaijiang, Dazhou 636250, China
The First People's Hospital of Jiangjin District, Chongqing 402260, China

Correspondence to: Min Xie (minxie@njnu.edu.cn) and Nan Wang (nan.wang@scu.edu.cn)

**Abstract.** Forest and vegetation fires are one of the major sources of air pollution and have triggered air quality issues in many regions of Asia. Measures to reduce fires may be a significant yet under-recognized option for effeciently improving air quality and averting the related premature deaths. Here we isolate the fire-specific fine particulate matter ($PM_{2.5}$) from monitoring concentrations using an observation-driven approach in the region. Fire-specific $PM_{2.5}$ concentrations average 2-15 µg/m³ during the fire season, with higher values in Southeast Asia (SEA), Northeast Asia (NEA), and northern India. The total $PM_{2.5}$ in Asia Pacific exhibits a rapid declining trend from 2014 to 2021, while fire-specific $PM_{2.5}$ decreases in early years but begins to reverse in SEA and NEA. The proportions of fire-specific $PM_{2.5}$ in NEA rises from 0.2 to 0.3 during the fire season, and in SEA increases from 0.2 in 2018 to 0.4 in 2021. Fire-specific $PM_{2.5}$ exposure caused 58,000 (95 % confidence interval (CI) of 32,600-82,600 ), 90,000 (95 % CI of 63,700-106,000 ), 157,000 (95 % CI of 110,000-186,000 ), and 29,300 (95 % CI of 18,000-39,700 ) premature deaths annually in SEA, East Asia (EA), Central Asia (CA), and NEA, respectively, accounting for 40.9% (95 % CI of 22.8%-57.7%), 14.9% (95% CI of 10.5%-17.6%) , 19.4% (95% CI of 13.5%-24.5%), and 24.1% (95% CI of 14.8%-32.5%) of numbers caused by the total $PM_{2.5}$. Analysis of infant mortality rate data and $PM_{2.5}$ exposure indicates that the total $PM_{2.5}$ exposure impacted more in richer areas, while

fire-specific PM$_{2.5}$ exposure affected more populations in poorer regions. Based on the positive
correlation between vapor pressure deficit and fire-specific PM$_{2.5}$, this study suggests that without
further regulation and policy intervention, the emerging growth trend of fire-specific PM$_{2.5}$ in Asia
Pacific is likely to continue under the influence of future climate change.
**1 Introduction**

Fine particulate matter (PM$_{2.5}$) is a complex mixture of anthropogenic and natural sources,

and has been the world's leading environmental health risk factor (McDuffie et al., 2021).
Observations show that emissions from forest and vegetation fires are one of the major sources of
PM and have triggered air quality issues in many regions (Reddington et al., 2021; Romanov et al.,
2022; Xie et al., 2022). Influenced by climate change, fires are becoming increasingly frequent
and destructive, and fire-specific PM$_{2.5}$ has begun to dominate the average annual PM$_{2.5}$ trends in
some areas (Burke et al., 2023; Wei et al., 2023). Compared with the direct exposure to flames and
heat of fires, exposure to fire smoke can affect much larger populations and pose significant public
health risks (Xu et al., 2023). The most severe public health impact of fire smoke on air pollution
comes from the generation of toxic PM. Recent studies suggest that fire-specific PM$_{2.5}$ may be
more influential than equal doses of ambient PM$_{2.5}$ (Xue et al., 2021; Aguilelra et al., 2023; Wei et
al., 2023). Exposure to fire-specific PM$_{2.5}$ can exacerbate a range of health problems, such as
premature mortality, cardiovascular and respiratory and other health issues (Aguilera et al., 2021;
Chen et al., 2021).

Studies have analyzed changes in fire-specific PM$_{2.5}$ concentrations and their health impacts

using chemical transport models, which are valuable for assessing conditions across different
locations and times (Reddington et al., 2021; Xue et al., 2021; Xu et al., 2023). Some studies focus
on individual fire events, defining fire influence by threshold values of biomass burning tracers
(e.g., PM$_{2.5}$ or CO) to identify fire-influenced measurements (Bytnerowicz et al., 2016; Landis et
al., 2018). Others use backward trajectory simulations to confirm fire influences but often
overlook smaller-scale fire emissions, which are harder to attribute. Accurately measuring
fire-specific PM$_{2.5}$ exposure is vital for assessing health and economic impacts, yet empirical
challenges persist. Recently, some studies have combined PM$_{2.5}$ observational data with fire
smoke observations to determine fire effects on air quality, that is trajectory-fire interception
method (TFIM) (Schneider et al., 2021; 2024). TFIM extracts unaffected time and spatial points,
employing statistical or machine learning techniques to estimate pollutant concentrations. This
data-driven approach does not depend on the fire emission databases that carry significant
uncertainties related to fuel type and location (Wiedinmyer et al., 2006; 2011; 2023), and
enhances reliability and timeliness, conserving computational resources while isolating
fire-specific air pollution (Aguilera et al., 2021; 2023).
Asia Pacific is one of the most densely populated regions in the world and faces severe air
pollution challenges (CCAC, 2024). Among the health risks associated with air pollution, Asia
Pacific has accounted for over 70% of global deaths attributed to air pollution (Lelieveld et al.,
2015; 2020; Giannadaki et al., 2018). Fire actively in the Northeast Asia (NEA) region has
recently become more extensive and is expected to continue escalating in the future due to climate
change (Huang et al., 2024; Gui et al., 2024). Fires in equatorial Southeast Asia (SEA) are
severely impacted by droughts induced by the El Niño-Southern Oscillation (Yin et al., 2020;
Zheng et al., 2023). South Asian are among the most vulnerable globally to the impacts of climate
change, which has increased the incidence of fire in Central Asia (CA). In addition to climate and
natural factors, the frequencies and sizes of fires are also largely human influenced through land
management practices in Asia Pacific. In East Asia (EA) and SEA, fires are used as agricultural
management tools, such as to remove agricultural residues and weeds, as well as for forest
clearance for agricultural purposes (Biswas et al., 2015; Phairuang et al., 2017). Fire activity in
Asia Pacific may release large amounts of smoke and harmful gases, leading to elevated
concentrations of air pollutants and negatively affecting human health and the environment
(Reddington et al., 2021). The fire-specific air pollution in Asia Pacific not only poses a threat to
the health of local residents but can also influence neighboring areas and even more distant
locations through atmospheric transport (Zhu et al., 2016; Qin et al., 2024; Du et al., 2024).
However, large disparities in geographic patterns exist in fire-specific air pollution and
population exposure researches, with related studies most centralized in high-income economies,
like North America and Europe (Aguilera et al., 2021; Tornevi et al., 2021; Korsiak et al., 2022;
Wei et al., 2023). In contrast, the world's most widely burnt regions, including the Asia Pacific,
remain underrepresented in literature due to resource inequality and inadequate funding (Petersen,
2021; Lin et al., 2024). On one hand, a major challenge to conduct researches on fire-related $PM_{2.5}$
pollution and population exposure is how to isolate the fire-specific $PM_{2.5}$ from observed
background levels. More than 70% of studies on fire-related datasets are concentrated in North
America and Europe, using various approaches such as chemical transport models, satellite-based
fire smoke plume analysis and statistical approaches to quantify fire-specific $PM_{2.5}$ (Aguilera et al.,
2021; Schneider et al., 2021; Korsiak et al., 2022; Wei et al., 2023; Lin et al., 2024). However,
there is still a lack of fire-specific $PM_{2.5}$ in many other regions, including Asia Pacific, which
accounts for 7.4% of the global burnt area and 27% of global cropland fires (Xu et al., 2023). on
the other hand, associated with the socioeconomic factors, increasing evidence highlights the
unequal distribution of exposure to and impacts of air pollution, attributed to the disparities in the
implement of measures, effectiveness of regulations, the adoption of clean energy technologies,
and differences in infrastructure and healthcare conditions (Tessum et al., 2019; Jbaily et al., 2022;
Kodros et al., 2022; Southerland et al., 2022; Rentschler and Leonova, 2023). However few
studies have focused on how fire-specific $PM_{2.5}$ exposure manifests along lines of inequality,
thereby exacerbating health disparities. Notably, there is a lack of research focusing on
contributions of fires activities to $PM_{2.5}$ in Asia Pacific, as well as the health and socioeconomic
impact of fire-specific $PM_{2.5}$.
This study utilized a trajectory-fire interception method (TFIM), and spatial-temporal interpolations
through machine learning algorithm to isolate fire-specific $PM_{2.5}$ from monitoring observations in Asia
Pacific. With the fire-specific $PM_{2.5}$, variations in contributions of fire activities to $PM_{2.5}$ in the
Asia Pacific are analyzed. The health impacts caused by fire-specific $PM_{2.5}$, and the relationship
between poverty levels and fire-specific $PM_{2.5}$ exposure in Asia Pacific were also examined.
Based on the climate factors related to fire activities, this study aims to demonstrate whether the
changing trends of fire-specific $PM_{2.5}$ will go on due to climate change.
**2 Data and Methods**
**2.1 Data**
2.1.1 Air quality Data
The continuous air quality observation data were obtained from the OpenAQ website
(http://openaq.org/), while data for the China region primarily comes from the Chinese National
Environmental Monitoring Center (http://www.cnemc.cn/en/). The total $PM_{2.5}$ between 2014 and
2020 were measured using observation data from 1,810 monitoring stations (Figure 1) located
throughout the Asia Pacific (65-133°E, 5-55°N). Additionally, the CO measurements from these
monitoring stations were utilized to validate the definition of fire influence using the TFIM
method.
2.1.2 Fire Point Data
The location of fires were obtained from the Fire Information for Resource Management
System (FIRMS). Rrchived fire pixels from the Moderate Resolution Imaging Spectroradiometer
(MODIS) on the Aqua and Terra satellites for Asia Pacific from 2010 to 2021 were downloaded.
The standard fire products with a resolution of 1 km×1 km for each fire pixel were utilized. More
information about MODIS measurements can be found in Giglio et al. (2003) and Justice et al.

(2011).

2.1.3 Additional Variables
To estimate fire-specific $PM_{2.5}$ concentrations, the study firstly used spatial-temporal
interpolation approach to calculate counterfactual $PM_{2.5}$ that is in absence of fire smoke. The
spatial-temporal interpolation approach was realized based on a machine learning methods with
multiple potential explanatory variables, including aerosol optical depth (AOD) data,
meteorological data, land use data, and other auxiliary information.
For AOD data, the reliability of the MODIS products onboard the U.S. Terra and Aqua
satellites has been extensively validated (Lyapustin et al. 2018; Mhawish et al., 2019; Choi et al.,
2019; Huang et al., 2020; Jin et al., 2023 ). The high resolution AOD product, with a resolution of
1 km, is derived using the Multi-Angle Implementation of Atmospheric Correction (MAIAC)
algorithm, which enhances the accuracy and spatial resolution of the AOD product (Lyapustin et
al., 2018). The MAIAC AOD data has recently been widely applied to retrieve ground-level $PM_{2.5}$
concentrations (He et al., 2020; Li et al., 2020; Wei et al., 2023).
Satellite remote sensing offers uniform coverage, but satellite data is only feasible under
clear-sky conditions. MAIAC AOD contains large data gaps due to ubiquitous presence of clouds.
To fill spatial-temporal gaps of MAIAC AOD, this study also supplemented MERRA-2 AOD

products. MERRA-2 is the first global reanalysis dataset of the satellite era, provided by NASA's Modeling and Assimilation Data and Information Services Center. It assimilates ground-based aerosol observations, with a horizontal resolution of 0.625° × 0.5° and a temporal resolution of 1 hour (Gelaro et al., 2017). Studies have used MERRA-2 aerosol products to conduct in-depth researches on atmospheric environmental issues in Asia (Jia et al., 2019; Feng et al., 2020). Additionally, MERRA-2 provides 50 aerosol products, including AOD, surface black carbon mass concentration, surface organic carbon mass concentration, and surface dust mass concentration. This study utilizes MERRA-2 reanalysis aerosol products as input data for constructing the AOD-$PM_{2.5}$ model.

Meteorological variables affect air pollution, therefore meteorological data provided by ERA5 reanalysis data serve as input factors for estimating the $PM_{2.5}$ in absence of fire smoke. ERA5 reanalysis data comes from ECMWF and assimilates as comprehensive observational data as possible (including ground observations, soundings, aircraft data, satellite observations, etc.). It is widely used in weather and climate-related research, with a horizontal resolution of 0.25° × 0.25° and divided into 37 vertical layers, with a resolution of 25 hPa from 750 to 1000 hPa and 50 hPa from 750 to 250 hPa, and a temporal resolution of 1 hour. The data used in the study included surface air pressure, 10-meter U and V wind fields, 2-meter temperature and dew point temperature, as well as specific humidity and temperature at 500 hPa and 850 hPa.

Land-use variables are proxies for emissions and background $PM_{2.5}$. In this study, the land-use coverage types collected from the MCD12Q1 Version 6 products, and the 16-day composite Normalized Difference Vegetation Index (NDVI) derived from MODIS were utilized as input factors for $PM_{2.5}$ estimation. In addition, the population counts obtained from LandScan was included to represent impact of human activities on air pollution. The gross domestic product (GDP) data are obtained from Wang and Sun (2023), measured in PPP 2005 international dollars.

Table 1 summarizes the original input features used in construction of machine learning method estimating fire-specific $PM_{2.5}$. Although the resolutions of different datasets in the machine learning method are quite distinct, the target data are spatially and temporally dispersed points. Therefore, the construction of machine learning method is essentially point-to-point. The input and output datasets are matched based on their relative positions, meaning that the input data are temporally and spatially closet to the output data.

Table 1. The original input features used in construction of machine learning method estimating

fire-specific $PM_{2.5}$

| Variation | Content | Spatial Resolution | Temporal Resolution | Source |
|---|---|---|---|---|
| $PM_{2.5}$ | $PM_{2.5}$ absent of fires | In situ | Hourly | OpenAQ, CNEMC |
| AOD | MAIAC AOD | 1km × 1km | Daily | MCD19A2 |
| Aerosol | 50 aerosol products | 0.62°× 0.5° | Hourly | MERRA-2 |
| Meteorology | surface air pressure 10m U and V wind fields 2m temperature 2m dew point temperature, specific humidity at 500 hPa and 850 hPa temperature at 500 hPa and 850 hPa | 0.25°× 0.25° | Hourly | ERA5 |
| Land use | Land coverage types | 500m × 500m | Yearly | MCD12Q1 |
| NDVI | Normalized difference vegetation index | 1km × 1km | Monthly | MOD13A3 |
| POP | Population counts | 1km × 1km | Yearly | LandScan |
| GDP | Gross domestic product | 1km × 1km | Yearly | Wang and Sun (2023) |

2.1.4 Health Data

To estimate the health impacts at a specific ambient $PM_{2.5}$ exposure, population data from LandScan and mortality rate data from the online Global Burden of Disease (GBD) database (http://ghdx.healthdata.org/gbd-results-tool) covering Asia Pacific from 2014 to 2020 were collected and used. The GBD database provides baseline mortality data for male and female populations across five-year age groups. This study considers health endpoints for four diseases: stroke (STROKE), chronic obstructive pulmonary disease (COPD), ischemic heart disease (IHD) and lung cancer (LC).

2.1.5 Infant Mortality Rates

The Infant Mortality Rates (IMR) dataset from NASA Socioeconomic Data and Applications Center was used as a proxy for population poverty levels in this study. The IMR is defined as the number of children who die before their first birthday for every 1000 live births in a given year (Barbier and Hochard, 2019; Reddington et al., 2021). IMR dataset has been widely used as poverty indicators, with specific thresholds to assess and categorized poverty levels ( Barlow et al., 2016; Barbier and Hochard, 2019). This study define population with IMR≤40 to be relatively not

poor, 41≤IMZ≤60 to be moderately poor, IMR≥61 to be relatively poor, which is similar to the
definition in Barbie and Hochard (2019).

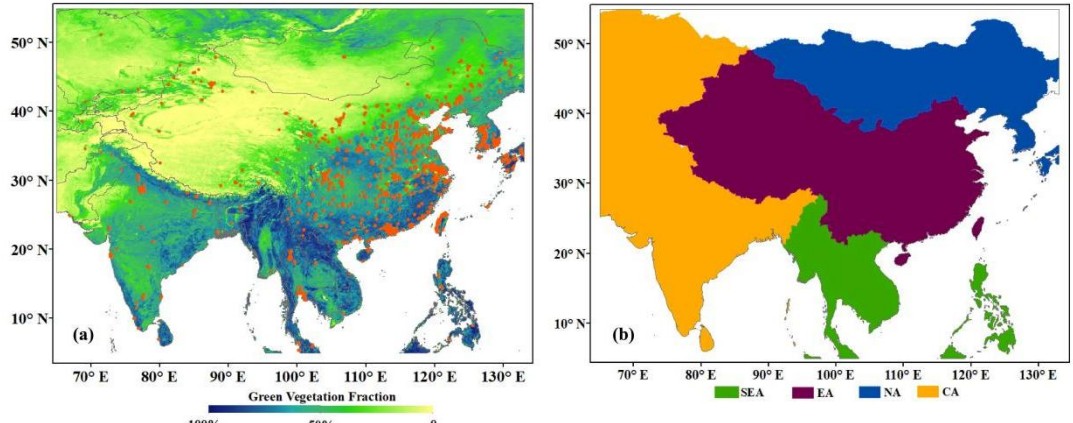


**Figure 1.** (a) Distribution of air quality monitoring stations in Asia Pacific, with shading color in
background indicating green vegetation fraction. (b) The specific areas of sub-regions including
Southeas Asia (SEA), East Asia (EA), Northeast Asia (NEA) and Central Asia (CA).

2.1.6 The Coupled Model Intercomparison Project Phase 6 data
Referring to previous researches, the positive relationship may exist between the vapor
pressure deficit (VPD) and the fire-specific $PM_{2.5}$ (Abatzoglou et al., 2016; Burke et al., 2023). To
validate this relationship and quantify the future trend of fire-specific $PM_{2.5}$ in Asia Pacific, VPD
was calculated using the projected temperature and relative humidity data from climate model
(GCM) ensembles under various emissions scenarios. The study examined VPD changes under
three commonly used climate scenarios (SSP1-2.6, SSP2-4.5, and SSP3-7.0) , based on monthly
data provided by 34 GCMs. To minimized uncertainty and account for internal variability, the
average VPD values for different regions in Asia Pacific were computed for each GCM and
emissions scenario.
**2.2 Methods**
2.2.1 Fire Influence definition
To understand how fire impact air quality, whether an ambient $PM_{2.5}$ measurement has been
influenced by fire should be determined. Following the TFIM method proposed by Schneider et al.
(2021), this study calculated the backward trajectories for monitoring stations over a 72-hour
period. The FLEXPART model (version 10.4), a Lagrangian particle dispersion model developed
by the Norwegian Institute for Air Research, was used for back-trajectories calculation.
FLEXPART v10.4 was driven using ERA5 reanalysis data at a temporal interval of 1 hour. These
trajectories were then spatially and temporally matched with fire hotspot data reported by FIRMS.
If the distance between the two was within 0.5°, an interception was considered to occur. If a
trajectory had more than the interception threshold, the $PM_{2.5}$ measurement at that time was
deemed to be influenced by fire. A schematic of the TFIM method is shown in Figure 2.

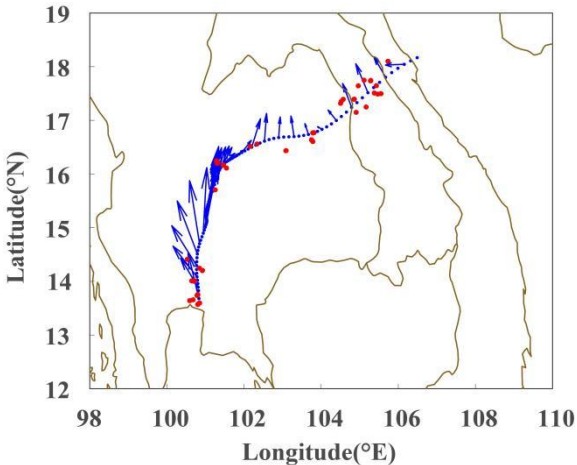


**Figure 2.** The schematic of trajectory-fire interception method (TFIM), where the blue lines
represent backward trajectories and red points indicate fire hotspots
2.2.2 Fire-specific $PM_{2.5}$ estimation
To estimate fire-specific $PM_{2.5}$ covering Asia Pacific from 2014 to 2020, the counterfactual
$PM_{2.5}$ unaffected by fire was interpolated through machine learning method, and then compared
with the ambient $PM_{2.5}$ measurement to get the fire-specific $PM_{2.5}$. The specific steps in Figure 3
were followed. Since there are no direct fire smoke observation data over Asia Pacific, the TFIM
method described in 2.2.1 was used as a substitute. First, using the TFIM method, the fire
influence periods for a given monitoring station time were determined. If a station experienced
over 6 hours of fire influence in a day, it was considered exposed to fire smoke on that day. Based
on the exposure definition, the station days exposed to fire were temporarily removed. Next, the
random forest method was employed to interpolate non-fire-affected $PM_{2.5}$ for all station days
categorized as fire-affected. Random forests are a combination of tree predictors, such that each
tree depends on the values of a random vector sampled independently and with the same
distribution for all trees in the forest (Breiman 2001). Since it is relatively robust to noise, random
forests are not prone to overfftting, so that it is carried in various fields of data mining (Lu et al.,
2021). In this study, we utilize random forest to estimate $PM_{2.5}$ that is absent of fire with multiple
input features. The algorithm provides insights into feature importance, allowing us to understand
which variables contribute most significantly to predictions. In our study, the feature importance
of 60 original input datasets (Table 1) were calculated based on random forest, and then $PM_{2.5}$
absent of fire was then estimated with the algorithm. This step provided background $PM_{2.5}$
estimation unrelated to fire contributions. The $PM_{2.5}$ from non-fire-affected station days was used
as the training, testing, and validation datasets to build the model, and interpolation estimation was
performed for background $PM_{2.5}$ for fire-affected station days. Finally, by subtracting the
non-fire-affected part from the ambient $PM_{2.5}$ measurement, the fire-specific $PM_{2.5}$ was estimated.

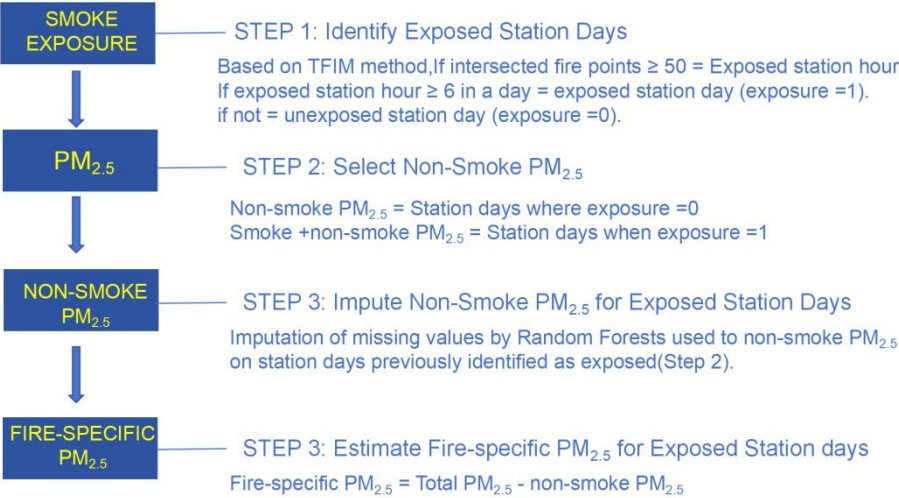


**Figure 3.** Flowchart of steps followed to estimate fire-specific $PM_{2.5}$

2.2.3 $PM_{2.5}$ health impact assessment

The disease burden attributable to $PM_{2.5}$ exposure was assessed using Health Impact Function

(HIF). The expression for this function is as follows:

$$\Delta Mort = B_i \times POP \times (1 - 1/RR_i)$$

where $\Delta Mort$ denotes the premature death due to $PM_{2.5}$ exposure for health endpoint i , $B_i$
represents mortality rate for endpoint i , POP is the exposed population, and $RR_i$ is the relative
risk associated with $PM_{2.5}$ exposure for health endpoint i .

With the advancement of epidemiological research, an Integrated Exposure-Response (IER)

equation integrates available RR information from multiple exposure-response functions,
including air pollution, active smoking, passive secondhand smoke exposure, and indoor cooking
fuel combustion scenarios. The IER equation combines findings from studies on both low and

high exposure concentrations to consider four major health endpoints (STROKE, COPD,IHD, and

LC). The expression for the IER has the following form:

$$RR = 1 + \alpha(1 - exp(-\gamma(C - C_0)^\delta))$$

Where C represents the PM$_{2.5}$ concentration, $C_0$ is the concentration threshold below which

health risks are negligible, and the parameters α、γ and δ represent the fitted parameters for health

endpoint i to describe the relative risk curve. The values for parameters can be found in studies by

Burnett et al. (2014) and Song et al. (2017). The values of these key parameters and their 95%

confidence intervals (CI) used in this study are also provided in Table S1.

## 3 Results

### 3.1 Estimating fire-specific PM$_{2.5}$

Fire hotspots number derived from the FIRMS products in Asia Pacific peaked during

February to April (with daily counts exceeding 1000), therefore we defined this period as fire

season in this study (Figure 4). In terms of spatial distribution, fire hotspots number in SEA is

more than double that of the other three regions during fire season. Fires in SEA mainly occur

during the pre-monsoon period (roughly February to April), due to widespread forest fires and

agricultural residues burning in preparation for planting before the arrival of the Asian summer

monsoon (Huang et al., 2017; Phairuang et al., 2017). The increase in fire activity coincides with

the establishment of stable temperature inversions over large areas of Thailand, Vietnam, Laos,

and southern China, while northern Thailand experiences hot, dry, and calm conditions that

facilitate the formation of haze (Reddington et al., 2021). Fire activities significantly decrease

after the onset of summer monsoon rainfall (in late April) and remain low until the beginning of

the dry season (in November). The fire occurrences in this region exhibit a certain degree of

interannual variability (Figures 4c and 4d), which is related to changes in atmospheric circulation

patterns, such as the India-Burma trough (Huang et al., 2017). In addition to climatic influences,

local fire management policies also play a role; for example, the implementation of stricter

agricultural burning policies in SEA mainland between 2016 and 2017 was associated with a

significant reduction in fire point counts. However, after 2018, the number of fire points once

again showed an upward trend.

Fire hotspots number in CA is slightly higher than EA during the fire season (Figures 4b and
4d). The dry and hot conditions before the monsoon in CA create favorable conditions for forest
fires in the dense vegetation of the Indian Peninsula. Additionally, the dry winter climate in CA
can also contribute to fire occurrences (Barik and Baidya, 2023). As a result, the peak fire point
counts in CA primarily occur in March-April and October-November. The climate conditions in
EA are complex. During spring and autumn, North China and Southwest China experience clear
weather, low precipitation, and dry vegetation, making them prone to forest fires, especially
during windy conditions. In the western Xinjiang region, the peak period for forest fires is
concentrated in the summer, particularly those caused by lightning, with a significant number
occurring in July-August. The NEA region is located relatively further north, with the start of the
growing season lagging behind the other three regions, while the end of the growing season occurs
earlier than in the other regions. As a result, the peak fire point period in NEA is delayed in spring
(March-May) compared to the other three regions, but slightly advanced in autumn. The average
daily number of fire points in CA, EA, and NEA has shown a slow increasing trend from 2014 to

2021.

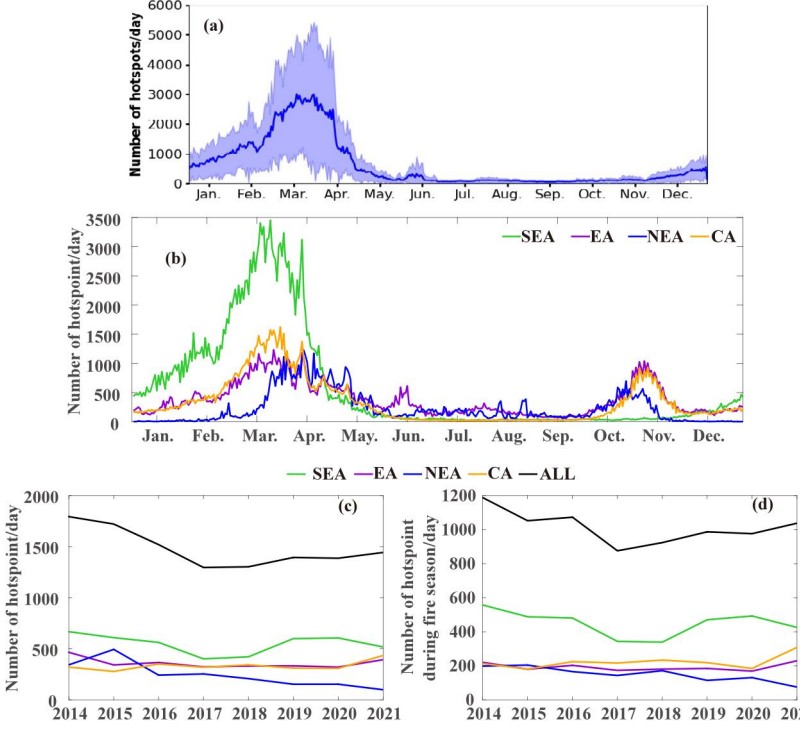


**Figure 4.** The variations from 2014 to 2021 of (a) day-to-day fire hotspots in Asia Pacific, (b)
day-to-day fire hotspots in four sub-regions, (c) annual averaged fire hotspots and (d) averaged
fire hotspots during fire season in different regions.      .

To isolate the fire-specific $PM_{2.5}$ based on TFIM, we should firstly justify the usability of

TFIM in the Asia Pacific, and then set a suitable threshold of fire hotspots interception for the

region. In this study, we select $PM_{2.5}$ as the fire emission tracer, as it is well know that $PM_{2.5}$ can

be emitted by fires. CO can also serve as a tracer for fire influence for CO can be produced from

incomplete combustion and has a long atmospheric lifetime. However, the range in CO is not as

large as it is for $PM_{2.5}$. The variations of $PM_{2.5}$ during high influence fires can be over 100 $\mu g/m^3$,

which is more than double that of clean period, while CO varies much milder. Besides, the much

more widespread $PM_{2.5}$ measurements compared to CO in Asia Pacific is another reason why

$PM_{2.5}$ is chosen as the tracer for fire emissions. We then compared the number of interception fire

hotspots with the measured $PM_{2.5}$ in Figure 5. In Figure 5a, correlation between the interception

number and $PM_{2.5}$ is not strong, indicating that identifying fire influence based on trajectory

interception of a single fire hotspot is not effective. When we set the interception threshold to 50,

the correlation significantly improves. This improvement may be due to larger and more fires

generating more $PM_{2.5}$. Figure 5c illustrates how the correlations varies as the interception

threshold changes. The correlation reaches it maximum at a threshold of 50. Therefore we set the

interception threshold to be 50 in measuring the fire influence on $PM_{2.5}$ in Asia Pacific. Compared

to the threshold of 20 in the North America proposed by Schneider et al. (2021), the interception

threshold in Asia Pacific is higher, because the study area is much larger and the relative smaller

scale of fires. This method eliminates fire hotspots that contribute minimally to $PM_{2.5}$ variations,

while including as many measurements as possible.

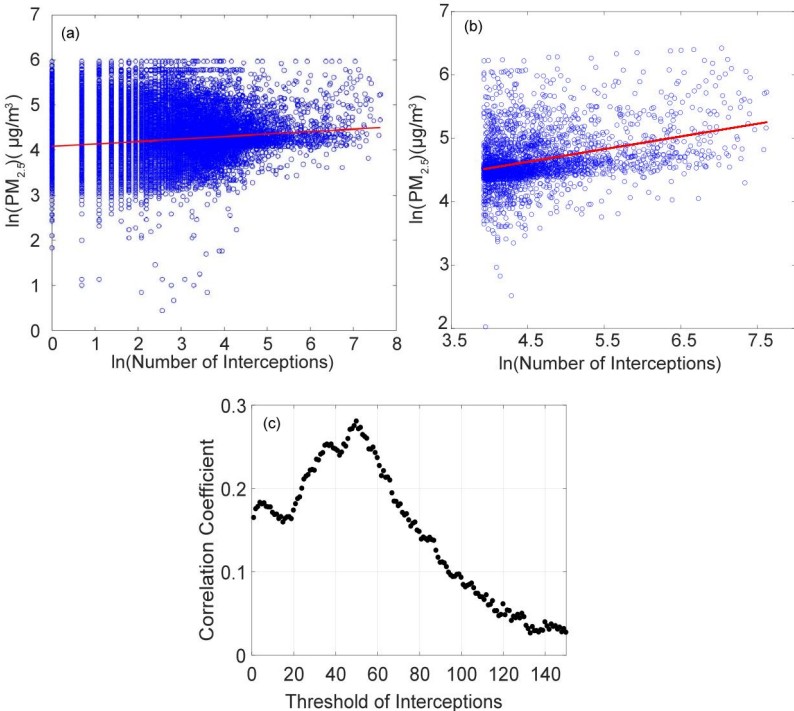

**Figure 5.** (a) and (b) scatter distributions of PM$_{2.5}$ concentrations against the number of fire hotspots when interception threshold is set to be 1 and 50, respectively. (c)correlation coefficient between PM$_{2.5}$ and the number of fire hotspots as a function of the interception threshold.

Using the TFIM method, we isolate the station days influenced by fires. To estimating the fire-specific PM$_{2.5}$, we employed a random forest model for interpolation to estimate the counterfactual PM$_{2.5}$ that is absence of fire influence, and then compare the PM$_{2.5}$ observation with the counterfactual PM$_{2.5}$ to get the fire-specific PM$_{2.5}$.

With muti-source data of station days that are absence of fires, we generate the datasets for machine learning model construction. There are totally 60 initial input variations, including 50 aerosol variables from MERRA2, MAIAC AOD, meteorological factors, land use, the NDVI and the GDP data. We ranked the importance of these variables using random forest, with the most 15 influential variables in Figure 6a. The most influential variables for PM$_{2.5}$ that are absence of fire is the surface black carbon mass (BCSMASS from MERRA2), followed by the surface mass concentrations of various PM$_{2.5}$ components, like organic carbon and dust. Meteorological factors contribute to explain variations in background PM$_{2.5}$. Temperature, pressure and humidity near ground can affect the formation of particles by influencing on chemical actions between precursors, while large-scale weather circulations also impact on pollutants transport and

accumulation through high level meteorological factors. In addition, other variations such as GDP
and NDVI also play a role in calculating background PM$_{2.5}$. GDP is expected to reflect the
economic conditions and background anthropogenic emissions among various regions, while
NDVI represents the vegetation cover status, which not only reflects the vegetation emissions but
also indicates the interception and deposition of PM$_{2.5}$ by vegetation. It is indeed important to
acknowledge the significant role of anthropogenic emissions in ambient PM$_{2.5}$ concentrations
across Asian countries. To comprehensively account for anthropogenic aerosols in this study, we
considered not only indirect reflection features, such as GDP and population, during the
construction of machine learning model, but also various aerosol data that directly reflect
anthropogenic sources. This includes black carbon, organic carbon, SO$_2$ surface mass
concentrations and so on. These data are derived from the MERRA-2 reanalysis, which
assimilates multiple aerosol remote sensing, emissions, and meteorological datasets using the
Goddard Earth Observing System Model. With these advances, MERRA-2 aerosol products can
provide reliable anthropogenic and natural aerosols (like dust). We then established an estimation
model using random forest with the 15 most influential input data to calculate the PM$_{2.5}$ that is
absence of fire. The background PM$_{2.5}$ estimates derived from the model were compared with
observations, with an estimating R$^2$ of 0.8958 and RMSE of 0.3370 μg/m$^3$ (Figure 6b). A little
under-estimation of the background PM$_{2.5}$ as it shows, the estimation has been highly correlated
with observations compared with the similar studies (Aguilera et al., 2021; 2023; Wei et al.,

2023).

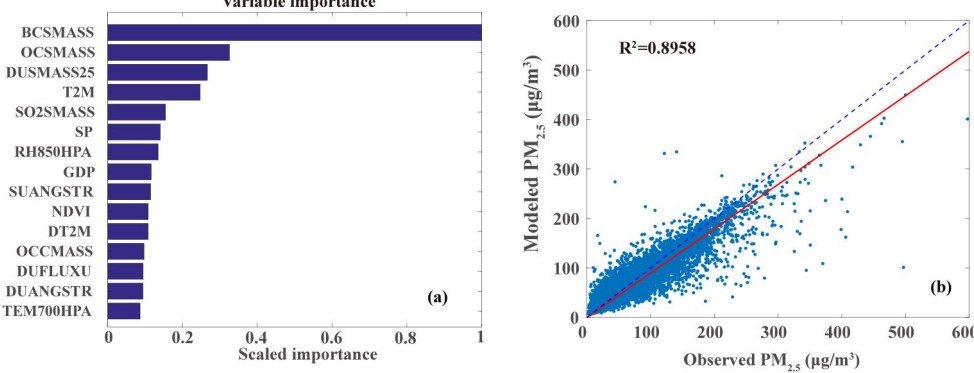


**Figure 6.** (a) Variation importance for the top 15 variables in estimating background PM$_{2.5}$; (b)
Scatter distribution between modeled and observed PM$_{2.5}$ that is absence of fire. Dashed blue lines
represents the reference, and red line is the linear model fit.

## 3.2 The spatial and temporal distributions of $PM_{2.5}$ and fire-specific $PM_{2.5}$

The fire-specific $PM_{2.5}$ was then estimated through subtracting the background $PM_{2.5}$ that is absence of fire from the monitoring $PM_{2.5}$. Figure 7a and 7b show spatial distributions of the 8-year mean total $PM_{2.5}$ and fire-specific $PM_{2.5}$ in Asia Pacific, respectively. $PM_{2.5}$ in Asia Pacific mostly has exceeded the health concentration standards for $PM_{2.5}$ set by the WHO (annual average not exceeding 10 µg/m³). The highest mean concentrations for total $PM_{2.5}$ are observed in northern India and Pakistan, followed by the Northeastern China, Indochina Peninsula, Mongolia and central India. To improve air quality, various measurements and particulate matter environmental standards have been implemented in countries of Asia Pacific, such as China's 'Air Pollution Prevention and Control Action Plan' since 2013, South Korea's enacting of the special act on the reduction and management of fine dust in 2018, India's launching of the National Clean Air Programme in 2019 and Thailand's amending the Enhancement and Conservation of National Environmental Quality Act in 2018, and so on. From 2014 to 2021, observed $PM_{2.5}$ concentrations saw substantial decrease in various regions of Asia Pacific (Figure 9). The highest $PM_{2.5}$ was monitored in EA during early period, but since 2018 $PM_{2.5}$ in CA began to exceed that of EA. In contrast, NEA and SEA have experienced lower annual average $PM_{2.5}$ concentrations.

The spatial distribution of fire-specific $PM_{2.5}$ is quite different with total $PM_{2.5}$, with highest concentrations appearing in SEA and Mongolia. As shown in Figure 4, fire hotspots number in SEA is more than twice as much as in other regions, which may partly explain the higher fire-specific $PM_{2.5}$ in this region. Mongolia has a large area of semi-arid forests with grass understories. Forests those located in mid to high latitude areas and dominated by a few coniferous tree species, are prone to a series of fire behaviors during droughts. Due to limited funding, firefighting efforts for forest fires in Mongolia are somewhat limited, leading to large-scale, long-duration forest and grassland fires during the dry season. Climate change, especially droughts, has intensified fire activities in Southern Siberia (including Mongolia), leading to a notable increase in fire numbers and shorter fire intervals (Hessl et al., 2016; Huang et al., 2024; Gui et al., 2024). As a result, higher fire-specific $PM_{2.5}$ can be found in the region of Asia Pacific. Besides, northern India is susceptible to fires before the monsoon and during the dry winter season, and northeastern and southwestern China are prone to forest fires in spring and autumn.

The annual average concentration of fire-specific PM$_{2.5}$ ranges from 2 to 8 µg/m³, surging to
between 2 and 15 µg/m³ during the fire season. Areas where the concentration of fire-specific
PM$_{2.5}$ surpasses 10 µg/m³ encompass northern India, the northeastern and southwestern China, as
well as several countries across SEA during fire seasons, as depicted in Figure 7 and 8. The values
for each region in Figure 8 are derived from the average values for sites within the region. In areas
with sparse stations (like Mongolia and Tibetan Plateau in Figure 1 ), while the calculation results
may not accurately reflect the fine spatial distribution within the region, using these averages to
represent the regional mean is still relatively reasonable.Contrary to the distribution of total PM$_{2.5}$,
fire-specific PM$_{2.5}$ is notably higher in NEA an SEA both in terms of annual average and during
the fire season. In addition, fire-specific PM$_{2.5}$ saw an increase trend in NEA since 2016, and in
SEA since 2018, with this trend more pronounced during the fire season. In contrast, fire-specific
PM$_{2.5}$ in EA and CA show slow decline. The total PM$_{2.5}$ has seen a significant decline thanks to
efforts in controlling anthropogenic emissions from industry and transportation. However,
fire-specific PM$_{2.5}$ decreases more slowly or even rebounds, leading to a gradual increase in the
proportion of fire-specific PM$_{2.5}$ within total concentrations. In NEA, the proportion during the
fire season has grown from 0.2 to 0.3, while in SEA it has risen from 0.2 in 2018 to 0.4 in 2021.
Proportions of fire-specific PM$_{2.5}$ in Malaysia, Cambodia and Brunei even exceeded 0.5 during the
fire season. (Figure 8). The proportions in the EA and CA also display gradual upward trends.

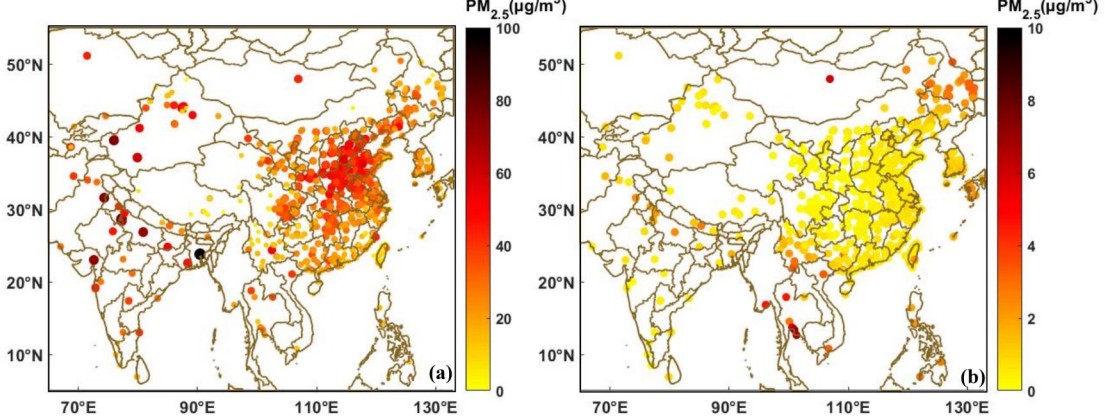


**Figure 7.** Distributions of (a) Mean PM$_{2.5}$ from all sources; (b) Mean fire-specific PM$_{2.5}$.

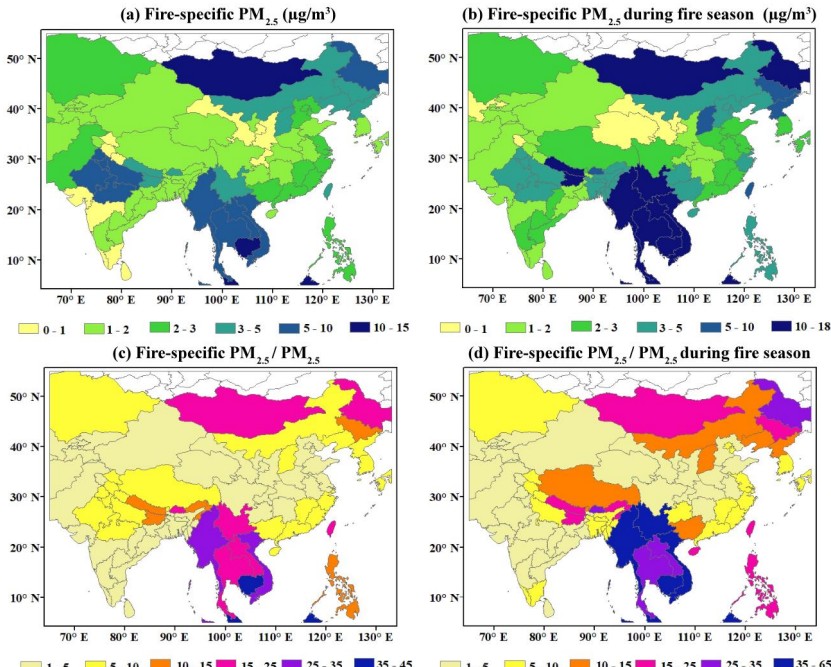

Figure 8. Regional averaged distributions of (a) annual mean and (b) fire season mean fire-specific PM$_{2.5}$; Proportion of (c) annual mean and (d) fire season mean fire-specific PM$_{2.5}$ to total PM$_{2.5}$. The values for each region in Figure 8 are derived from the average values for sites within the region.

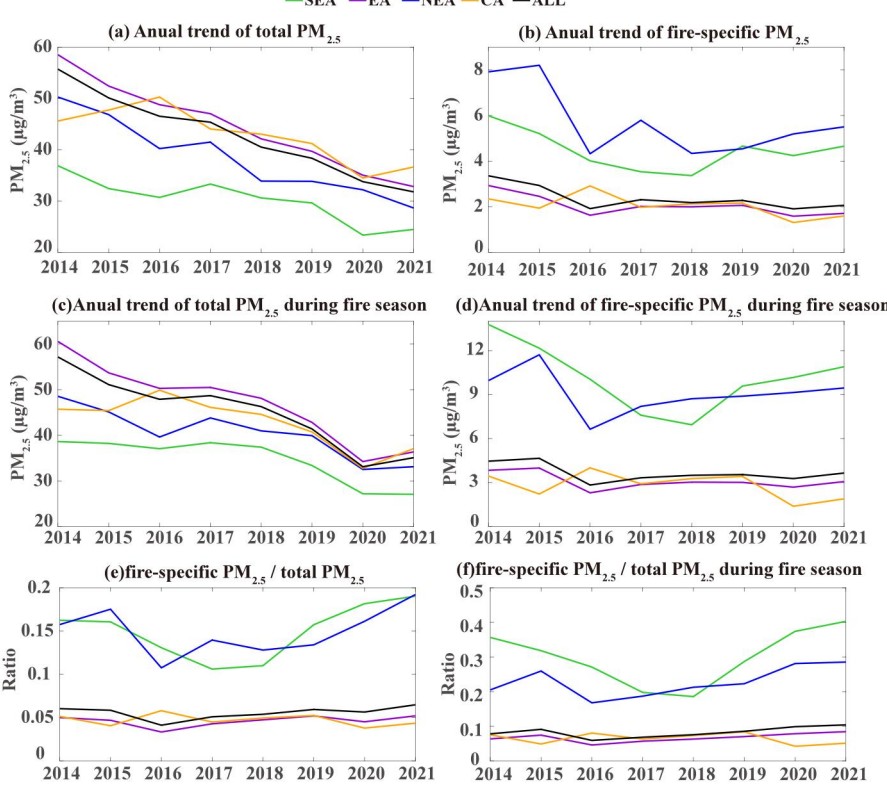

Figure 9. Temporal variations of (a) annual mean PM$_{2.5}$ and (b) fire season mean PM$_{2.5}$ in different regions; (c)(d) similar to (a)(b), but for fire-specific PM$_{2.5}$; (e)(f) similar to (a)(b), but for proportions of fire-specific PM$_{2.5}$ to total PM$_{2.5}$.

3.3 The fire-specific $PM_{2.5}$ exposure and health impact

To illustrate the population exposure, we then calculated the population-weighted $PM_{2.5}$ and

fire-specific $PM_{2.5}$ from 2014 to 2021 (Figure S1). Population-weighted $PM_{2.5}$ in different regions
saw a significant decline during the 8 years, with reductions of 30.5% in SEA, 41.1% in EA,
31.4% in NEA and 7.9% for CA, amounting to an overall decrease of 39.9% for the entire region.
$PM_{2.5}$ concentrations are high in densely populated areas of CA, such as northern India,
Bangladesh, and Pakistan (Figure S2), resulting in higher population-weighted $PM_{2.5}$. This
indicates that population in CA is more likely to be exposed to $PM_{2.5}$. In EA, population-weighted
$PM_{2.5}$ concentrations are higher in the east and lower in the west, which is consistent with the
distribution of population density in the region. The distributions of population-weighted $PM_{2.5}$ in
SEA and NEA are similar to their averaged $PM_{2.5}$. During fire seasons, distributions of population
exposure to $PM_{2.5}$ differ from those of total $PM_{2.5}$. Population-weighted fire-specific $PM_{2.5}$ in SEA
is higher than mean $PM_{2.5}$, indicating populations in SEA is more vulnerable to fire-specific $PM_{2.5}$
exposure. However, population-weighted $PM_{2.5}$ in CA is slightly lower than mean $PM_{2.5}$.

We then estimated the averted premature deaths due to changes in exposure to $PM_{2.5}$ from

eliminating fire emissions. Eliminating fire-specific $PM_{2.5}$ can avert approximately 58,000 (95 %
CI of 32,600-82,600 ) premature deaths annually in SEA, 90,000 (95 % CI of 63,700-106,000 ) in
EA, 157,000 (95 % CI of 110,000-186,000 ) in CA and 29,300 (95 % CI of 18,000-39,700 ) in
NEA. These account for about 40.9% (95 % CI of 22.8%-57.7%), 14.9% (95% CI of
10.5%-17.6%), 19.4% (95% CI of 13.5%-24.5%), and 24.1% (95% CI of 14.8%-32.5%) of the
total annual premature deaths attributed to $PM_{2.5}$. During fire season, these proportions can rise to
57.7% (95% CI of 27.3%-81.6% ), 19.5% (95% CI of 12.3%-24.6% ), 21.6% (95% CI of
14.8%-27.4% ), and 31.6% (95% CI of 17.2%-44.4% ). Distributions of premature deaths due to
$PM_{2.5}$ in CA and NEA (Figure 10) are closely aligned with population distribution (Figure S2),
because in these regions areas with higher population density tend to expose in higher $PM_{2.5}$. The
highest number of premature deaths attributed to fire-specific $PM_{2.5}$ occur in Myanmar, Vietnam,
northern India, and Pakistan, with notable increases during the fire season in Thailand and
southwestern China. Distributions of premature deaths attributed to $PM_{2.5}$ relative to regional
population proportions closely resembles the $PM_{2.5}$ distribution, with areas exceeding 50 per
100,000 mainly located in regions where annual mean PM$_{2.5}$ exceeds 40 µg/m³. Similarly, the
distribution of premature deaths caused by fire-specific PM$_{2.5}$ aligns closely with PM$_{2.5}$
distribution (Figure 10d), with areas exceeding 20 per 100,000 predominantly found in the
fire-prone Southeast Asian Peninsula, Mongolia, and northeastern China. The number of annual
premature deaths due to fire-specific PM$_{2.5}$ in the whole study region is around 1.7 million,
accounting for 47.2 per 100,000 of the total population.

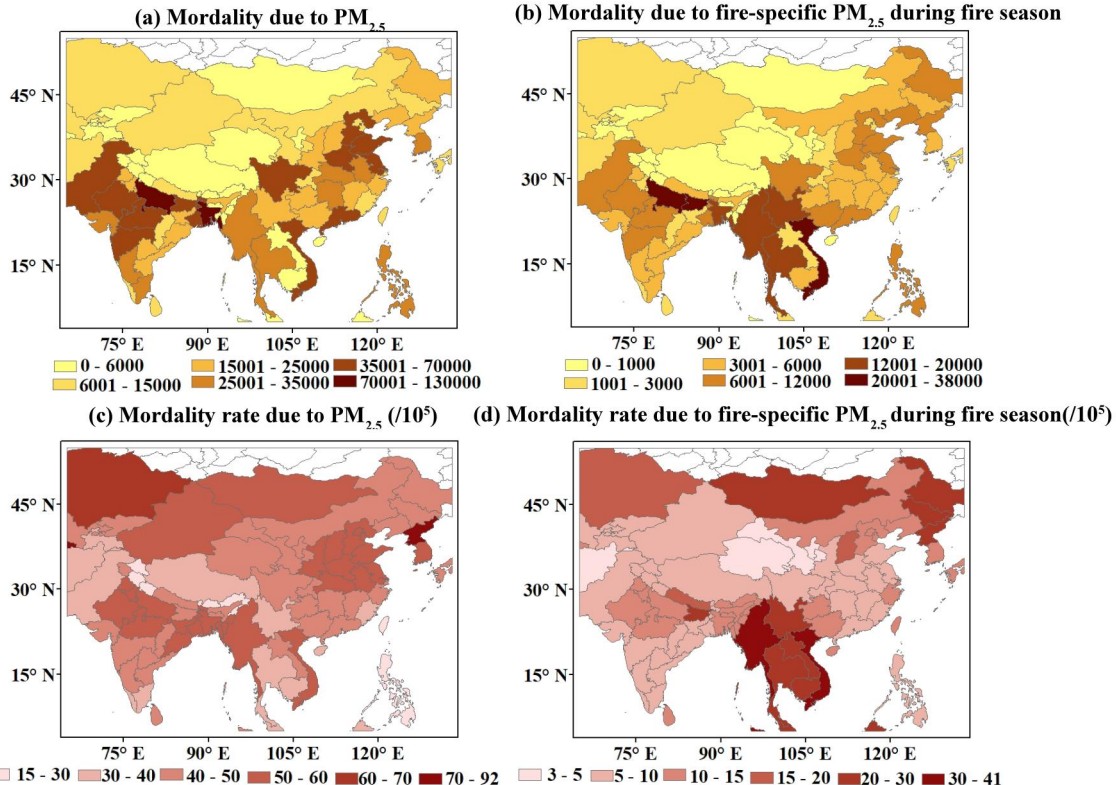


**Figure 10.** Distribution of premature deaths numbers due to (a) PM$_{2.5}$ and (b) fire-specific PM$_{2.5}$,
and the proportion of premature deaths relative to the local populations due to (c) PM$_{2.5}$ and (d)
fire-specific PM$_{2.5}$.
We further examined the poverty levels of Asia Pacific's population exposed to PM$_{2.5}$.
Figure 11 illustrates total PM$_{2.5}$ and fire-specific PM$_{2.5}$ plotted against poverty proxy (IMR) data
in Asia Pacific. For total PM$_{2.5}$, regions with IMR ≤ 60 show a gradual decrease in PM$_{2.5}$
exposure levels as IMR values increase. In low IMR areas (IMR ≤ 10), the average PM$_{2.5}$ (44.2
µg/m³) is significantly higher than that in regions with relatively higher IMR (41 ≤ IMR ≤ 60),
where the PM$_{2.5}$ averages at 28.3 µg/m³. In high IMR areas (IMR ≥ 61), the PM$_{2.5}$ exposure level
increases again to 37.0 µg/m³. While for fire-specific PM$_{2.5}$ the trend is reversed, with higher IMR
regions (IMR ≥ 40) are exposed to higher PM$_{2.5}$, while lower IMR regions (IMR < 40)
experience relatively lower PM$_{2.5}$. During fire season, populations in regions with IMR $\geq$ 41 and
≤60 are exposed to the highest fire-specific PM$_{2.5}$.
It is found that populations in "not poor" areas (IMR < 40) are exposed to higher mean PM$_{2.5}$
from all sources, but lower fire-specific PM$_{2.5}$. This indicates that PM$_{2.5}$ pollution during the study
period is primarily driven by economic and urban development. Conversely, "moderately poor"
populations (41 $\leq$ IMR $\leq$ 60) experience lower total PM$_{2.5}$ exposure, but higher fire-specific
PM$_{2.5}$ exposure. In "very poor" areas (IMR $\geq$ 61), both total PM$_{2.5}$ and fire-specific PM$_{2.5}$ are
high, making populations in these areas more susceptible to health impact of PM$_{2.5}$.

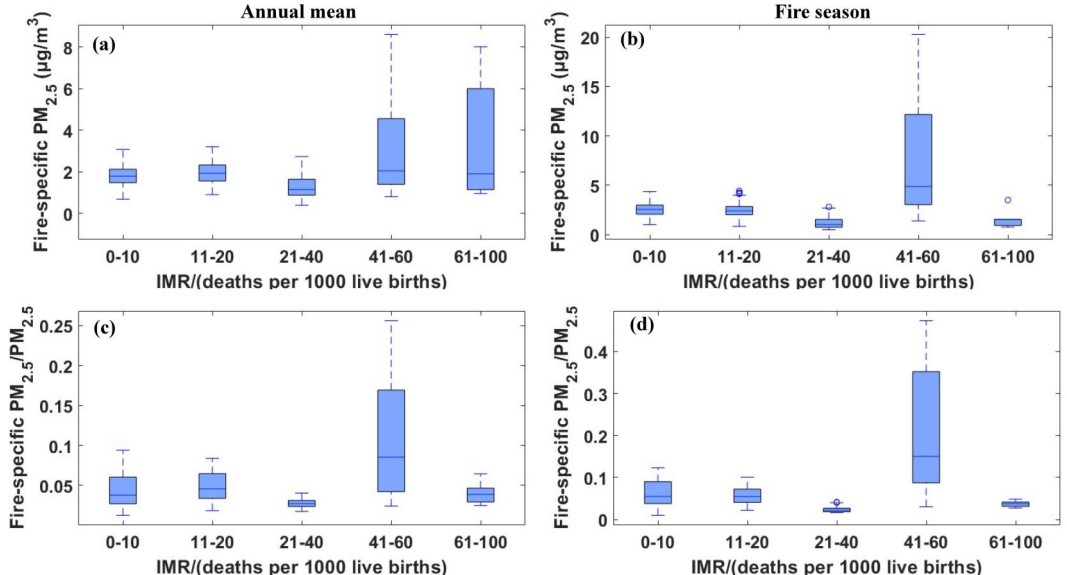


**Figure 11.** Annual mean (a) total PM$_{2.5}$ and (c) fire-specific PM$_{2.5}$ versus binned infant mortality
rate (IMR) values across the Asia Pacific. (b) (d) are similar to (a)(c), but for fire season mean.
3.4 Future trends of fire-specific PM$_{2.5}$ under climate change
Previous analysis indicates that fire-specific PM$_{2.5}$ in different regions have rebounded to
some extent, with more significant increase in SEA and NEA. Whether this trend will continue or
be altered by occasional climate conditions is uncertain. Many studies have attempted to
understand the climate drivers of increased fire activities and how these factors may change in the
future (Abatzoglou and Williams, 2016; Xie et al., 2022; Barik and Baidya, 2023; Burke et al.,
2023; Gui et al., 2024). These studies provide strong evidence that interannual variations in
climate factors are drivers of fire activities and changes in fire-specific PM$_{2.5}$. Based on future
change of these climate drivers predicted by GCMs, assuming no intervention, fire activities may
increase with global warming. With numerical model simulation, researches reveal that
fire-specific PM$_{2.5}$ will see rise in the future. To corroborate the future changes in fire-specific
PM$_{2.5}$ of Asia Pacific, we calculated mean VPD during fire season for different regions, and relate
these values to fire-specific PM$_{2.5}$. It is obvious that VPD is positively related to log of
fire-specific PM$_{2.5}$ (Figure 13a). Climate drivers can explain 35% of fire-specific PM$_{2.5}$ variations
in Asia Pacific, with variation in CA most sensitive to VPD (65%). The multi-model ensemble
mean of 34 GCM projections indicates a future increasing trend in VPD, with a pronounced rise in
SEA, followed by EA and CA, while the increase is weaker in NEA. These results suggest that the
emerging growth trend of fire-specific PM$_{2.5}$ in Asia Pacific is likely to continue under the
influence of future climate change. For more dynamic and spatially detailed characteristics, more
data will have to be integrated into modelling calculations to better understand the evolution of
fire occurrences and pollutants release under future climate impacts.

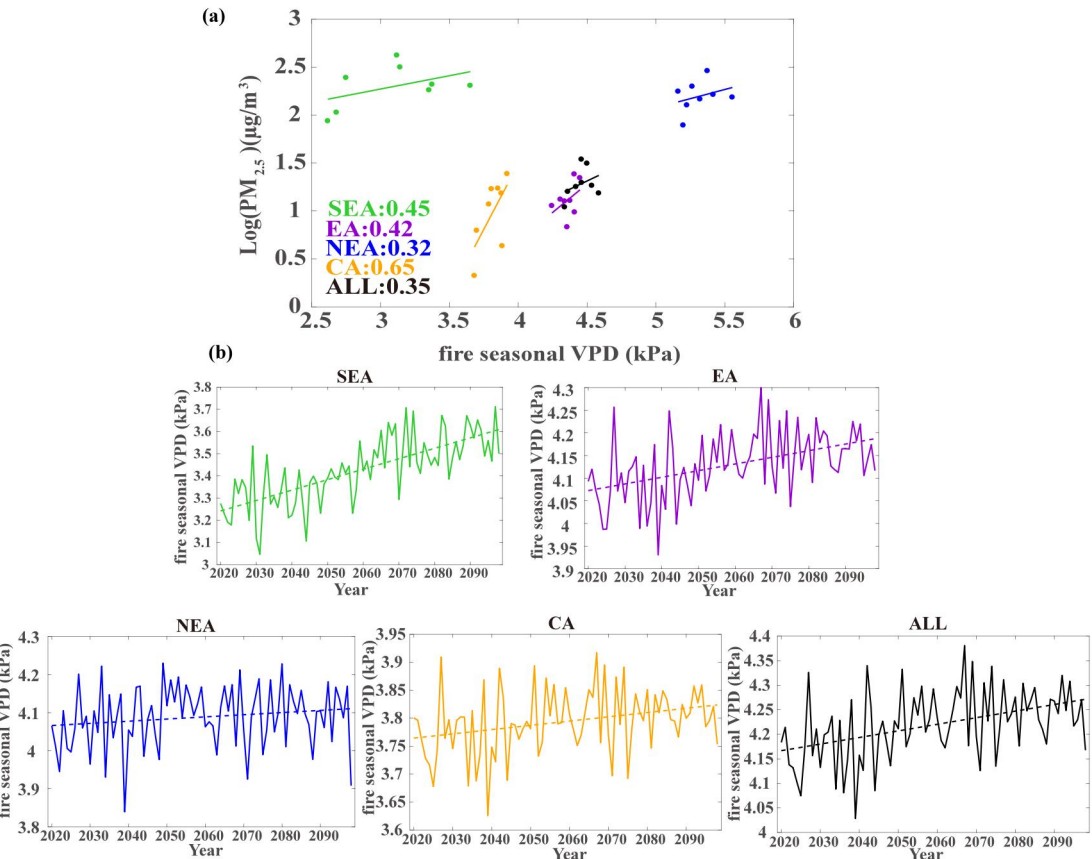

**Figure 12.** (a) Interannual variations of vaper pressure deficit (VPD) versus the log of averaged
fire-specific PM$_{2.5}$ during fire season; (b) future VPD derived from multi-model ensemble mean of
34 GCM projections

**4 Conclusion and discussion**

In this study, we explored the contribution of forest and vegetation fires to air quality and public health across the Asia Pacific. We isolate fire-specific $PM_{2.5}$ from the monitoring data for Asia Pacific using TFIM and spatiotemporal interpolation in this study. One advantage of this dataset is that it is driven by monitoring concentrations rather than relying on emission databases, which may probably ignore contributions of pollutants from smaller-scale fire emissions, and carry considerable uncertainty, especially with the evident underestimation of agricultural fire emissions. Moreover, this method offers reliability and timeliness, effectively saving computational resources and storage space for isolating fire-related air pollution.

Our analysis reveals geographical disparities in population exposure to $PM_{2.5}$ and fire-related air pollution in Asia Pacific. Thanks to the the establishment of $PM_{2.5}$ air quality standards and pollution control measurement by countries, $PM_{2.5}$ population exposure saw an obvious declining trend from 2014 to 2021 in Asia Pacific, with population-weighted $PM_{2.5}$ in 2021 reduced by 39.9% compared to 2014. High $PM_{2.5}$ concentrations are observed in EA and CA, concentrated in densely populated areas, leading to substantially higher population-weighted concentrations than mean $PM_{2.5}$. In contrast, fire-specific $PM_{2.5}$ decreased in the early years but began to reverse recently in Asia Pacific. SEA and NEA experienced the most obvious increase in fire-specific $PM_{2.5}$ in recent years, while EA and CA saw a slight increase. As a result, a gradual increase in the proportion of fire-specific $PM_{2.5}$ within total concentrations can be observed.

We found that fire-related $PM_{2.5}$ could pose a significant public health threat in Asia Pacific, contributing to approximately 334,300 (95 % CI of 224,000-414,000) premature deaths each year. The annual disease burden due to $PM_{2.5}$ exposure can be reduced by 40.9% (95 % CI of 22.8%-57.7%), 14.9% (95% CI of 10.5%-17.6%), 19.4% (95% CI of 13.5%-24.5%), and 24.1% (95% CI of 14.8%-32.5%) in SEA, EA, CA, and NEA, respectively, averting 58,000 (95 % CI of 32,600-82,600 ), 90,000 (95 % CI of 63,700-106,000 ), 157,000 (95 % CI of 110,000-186,000 ) , and 29,300 (95 % CI of 18,000-39,700 ) premature deaths. It is important to note that our calculations do not account for the potentially higher toxicity of fire-specific $PM_{2.5}$ compared to other sources, which could lead to an even greater number of premature deaths and related illnesses. Using infant mortality rates as a poverty proxy, we found that populations in Asia

Pacific are disproportionately exposed to $PM_{2.5}$. Populations in "not poor" areas (IMR $\leq$ 40) are
exposed to higher total $PM_{2.5}$, while poor populations are more vulnerable to health impacts of
fire-specific $PM_{2.5}$. Our study indicates that the fire-related air pollution is also a serious issue in
many poverty areas, yet it receives less attention. This situation warrants further investigation to
explore the underlying causes and characteristics, ultimately providing more scientific evidence
for effective management strategies. Based on the positive correlation between VPD and
fire-specific $PM_{2.5}$, the study suggests that without further regulatory and policy intervention, the
emerging growth trend in fire-specific $PM_{2.5}$ in Asia Pacific is likely to continue under the
influence of future climate change.
Interestingly, the increasing trend in fire-specific $PM_{2.5}$ appears inconsistent with the
declining trend in the number of fire points in Asia Pacific. In earlier years, vegetation fires in the
region were dominated by agricultural fires, characterized with smaller-scale burning areas but
more fire point numbers. Countries have implemented various policies to reduce agricultural fires,
such as China's measures to minimize straw burning and Thailand's alternative energy
development plans, like zero-burning policy. The enforcement of these policies has, to some
extent, reduced fire point numbers and emissions from agricultural fires in Asia Pacific (Kumar et
al., 2020; Panda et al., 2023). However, fire emissions in the region are also influenced by wildfire
emissions related to climate change. Wildfires usually occur in natural vegetation and are
characterized by larger-scale burning areas that are more challenging to extinguish (Gui et al.,
2024; Huang et al., 2024). As a result, the emissions per unit of biomass burned in wildfires far
exceed those from agricultural fires (Reddington et al., 2021; Jones et al., 2024). In this study, we
analyzed historical data and found the positive relationship existing between VPD and
fire-specific $PM_{2.5}$ across different regions of Asia Pacific. Based on this, we can roughly infer
future trend in fire-specific $PM_{2.5}$ through examining the VPD future trends, assuming that
relationship between future VPD and fire-specific $PM_{2.5}$ continues to exist. Of course, studying the
future trends of fire-specific $PM_{2.5}$ will require integrating more data and methods for a more
precise analysis, which is a direction for our future research. To explain the inconsistent of
changes in fire point numbers and emissions, it is proposed that increasing emissions from natural
wildfires driven by climate change are have contributed to the rise in fire-specific $PM_{2.5}$ in Asia
Pacific, although less fire points are found. This hypothesis may be further verified in the future
studies.
This study indicate that the contributions of fire-specific $PM_{2.5}$ to air quality and health
impact are becoming increasingly significant and deserve more attention when developing air
pollution standards and control measurements in Asia Pacific. These variations suggest that the
decreases in pollutant concentrations from traffic and industrial sources and the associated health
benefits may be offset by increases in pollutant concentrations from fires. Measures to reduce fires
may be a significant yet under-recognized option for effeciently improving air quality and averting
the related premature deaths.
**Data Available Statement**
Air quality observation data can be acquired from http://openaq.org/ and http://www.cnemc.cn/en/.
The ERA5 data can be respectively downloaded from
https://cds.climate.copernicus.eu/cdsapp#!/dataset/reanalysis-era5-pressure-levels. The fire point
data are available at https://earthdata.nasa.gov/firms. The health data can be accessed in
http://ghdx.healthdata.org/gbd-results-tool. The infant mortality rates data can be found at
https://www.earthdata.nasa.gov/data/catalog/sedac-ciesin-sedac-pmp-imr-v2.01-2.01. The
Coupled Model Intercomparison Project Phase 6 data can be get from
https://aims2.llnl.gov/search/cmip6/. The aerosol optical depth data are available at
https://www.earthdata.nasa.gov/data/catalog/lancemodis-mcd19a2n-6.1nrt and
https://disc.gsfc.nasa.gov/datasets?project=MERRA-2. The landuse data can be accessed in
https://lpdaac.usgs.gov/products/mcd12q1v006/. And the population data can be found at
https://landscan.ornl.gov/.
**Author contributions.** HL, MX and NW conceived the study, designed the experiments, conducted
the data isolation and prepared the initial draft manuscript. JJ, JY and KL collected the data and
assessed the health impacts of air pollution. HL, BL and BZ perform the analysis, engaged in
constructive discussions, reviewed and edited the manuscript. HL, MX and BL secured financial
support for the project leading to this publication. DM, MX, YZ and MW provided additional manuscript
reviews.
**Competing Interest:** The authors declare no conflict of interest.
**Financial support:** This work was supported by the National Natural Science Foundation of China
(42205186, 42275102), Chengdu Plain Urban Meteorology and Environment Observation and
Research Station of Sichuan Province open fund (CPUME202405), the Chongqing Natural Science
Foundation (cstc2021jcyj-msxmX1007, 2024NSCQ-KJFZMSX0258), Special Science and
Technology Innovation Program for Carbon Peak and Carbon Neutralization of Jiangsu Province
(BE2022612), the key technology research and development of Chongqing Meteorological Bureau
(YWJSGG-202215; YWJSGG-202303) and the research start-up fund for the talented person
recruitment of Nanjing Normal University (184080H201B57).

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
