# Peer review of "Figure S1. Distribution of air quality monitoring stations in Pacific Asia, with shading color in the background indicating green vegetation fraction."

_EGUsphere, 2025_

## Referee Comment (RC1)

This study explores the contributions of fires to air quality in Asia Pacific through isolating fire-specific $PM_{2.5}$ using data-driven method. A comparative analysis of total $PM_{2.5}$ and fire-specific $PM_{2.5}$ variations, along with their effects on public health is conducted. The obvious decline in total $PM_{2.5}$, coupled with a gradual increase in the proportion of fire-specific $PM_{2.5}$, suggests that measures to reduce fires may be a crucial yet under-recognized option for improving air quality and preventing related premature deaths in the future. The authors also reveal differences in spatial distributions of total and fire-specific $PM_{2.5}$. Total $PM_{2.5}$ is mainly concentrated in densely populated areas, while relatively high fire-specific $PM_{2.5}$ disproportionately affect poor populations. These conclusions indicated that the contributions of fires to air quality and health impact are becoming increasingly significant and warrant more attention when developing air pollution standards and control measurements in Asia Pacific. Overall, this study is well-structed, clearly presented, and the results are meaningful. However, some issues still need to be addressed before it can be published:

1. Line 28, Line 34: The full names of abbreviations should be given when their first appearance. What do "VPD" and "PM" stand for?

2. Figures: it is recommended to increase the resolution of the figures to enhance clarity and to enlarge the font sizes of annotations and icons within the figures.

3. Figure 1: Authors conducted the analysis with several sub-regions in the study, like EA, CA, ESA and NA. However, there are no figures or descriptions indicating the specific areas of each sub-region. It is necessary to add a sub-figure in Figure 1 providing this information.

4. Line 116-117: Please provide the references regarding the validation of the reliability of MODIS AOD, especially concerning the study regions in this manuscript.

5. In 2.2 Method: There appears to be a lack of information regarding how the authors align the resolutions of the input and output dataset of machine learning method,

considering that the resolutions of different dataset are quite distinct.

6. Figure 4: The line colors in Figure 4 (c) and (d) are somewhat difficult to distinguish, especially since the colors for CA and EA are quite similar, and the two datasets are also close in value, making it hard to interpret. Besides, the figure caption should provide more information about each sub-figures to make the figure stand out of the texts.

7. Line 294: GDP data are included in the machine learning method to estimate the counterfactual $PM_{2.5}$ that is absence of fire influence, however there is no any information describing the data source in the Section 2.

8. Figure 7 and Figure 8: The original isolated fire-specific $PM_{2.5}$ appears to be corresponded to site data shown in Figure 7. However, it is unclear how the authors calculate the fire-specific $PM_{2.5}$ values in each area presented in Figure 8?

9. Line 395-396: "The number of annual premature deaths due to $PM_{2.5}$ ....", does "$PM_{2.5}$" here mean fire-specific $PM_{2.5}$ or total $PM_{2.5}$?

10. Line 486-489: Authors should provide some evidence or references supporting the claim of decline in anthropogenic fire alongside increase in wildfires in Asia Pacific. If not, this statement should be omitted to maintain accuracy and credibility.

---

## Referee Comment (RC2)

**Review of "The contribution of fires to PM$_{2.5}$ and population exposure in Pacific Asia" by Lu et al.**

This study investigated the contribution of forest and vegetation fires to PM$_{2.5}$ and public health across Pacific Asia. The fire-specific PM$_{2.5}$ was effectively isolated from the monitoring data using the trajectory-fire interception method (TFIM) in combination with the random forest method. Based on the reliable and timely dataset, the geographical disparities in population exposure to both PM$_{2.5}$ and fire-specific PM$_{2.5}$ were estimated. The topic is novel, and the findings provide valuable insights for PM$_{2.5}$-related public health. Moreover, this manuscript is well-structured and readable. However, several issues need to be clarified before it can be considered for publication. Below are specific comments:

**Specific Comments**

1.  I suggest changing the regional abbreviations to Southeast Asia (SEA), Northeast Asia (NEA), East Asia (EA) and Central Asia (CA), respectively. And these key regions should be marked in Figure 1.

2.  To derive fire-specific PM$_{2.5}$ concentrations, a machine learning method was employed using aerosol variables, AOD, meteorological factors, land use, NDVI, and the GDP data. Ultimately, 15 of these variables were selected to fit PM$_{2.5}$ concentrations. In recent years, many Asian countries have implemented strict anthropogenic emissions reduction strategies to mitigate PM$_{2.5}$ pollution (see lines 322-330 for details). Anthropogenic emissions play crucial role in PM$_{2.5}$, but they are not incorporated into the machine learning model. While some variables, such as GDP and population, can indirectly reflect changes in anthropogenic emissions, the relationship needs to be further clarified. Therefore, the authors should briefly describe the machine learning method and variables used, even in the supporting information.

3.  Section 3 Results: It is recommended that additional subheadings (e.g., 3.1, 3.2, 3.3, ...) be added to improve clarity. Furthermore, more attention should be given to key regions (EAS, EA, NA and CA) during wildfires. Regions with extremely sparse stations (Figure 1), such as Mongolia and the Tibetan Plateau, may be masked out due to the large uncertainties in

their results.

4. Figure 10 and 11: To my knowledge, the Health Impact Function (HIF) exhibits large uncertainty due to the relative risk (RR). Additionally, the Integrated Exposure-Response (IER) equation varies by region and population, resulting in a confidence interval in the estimated number of premature deaths. Although these key parameters are from Burrett et al. (2014) and Song et al. (2017) (please note that the citation of Song is missing in the reference list) in this study, they should still be explicitly provided.

5. Figure 12: The results related to vaper pressure deficit (VPD) appear somewhat disconnected from the main body of this manuscript.

**Technical Corrections**

1. Normally, references are sorted alphabetically rather than chronologically.

2. Line 230: "in this study" instead of "is this study"

3. Figure 8: Please add the unit.

4. The manuscript contains minor spelling and grammatical errors that should be corrected.

---

## Referee Comment (RC3)

This study isolated the fire-specific $PM_{2.5}$ from monitoring concentrations through an observation-driven method, and provides insights into the evolving dynamics of fire-specific $PM_{2.5}$ in Pacific Asia. The observed decline in overall $PM_{2.5}$ levels juxtaposed with the rising proportion of fire-specific $PM_{2.5}$ presents a concerning trend, particularly as it suggests the shift in the dominant sources of emissions from anthropogenic activities to more unpredictable fires. This shift not only complicates air quality management efforts but also disproportionately affects vulnerable populations, exacerbating public health risks. Based on the positive correlation between vapor pressure deficit and fire-specific $PM_{2.5}$ highlight the potential exacerbating effects of climate change on future air quality. Overall, this assessment is valuable to call for more attention and researches in the complex interplay of fire-specific air pollution, public health and climate change in the region. However, some issues still need to be improved:

(1) Line 28: Please provide the full name of VPD when it appears the first time in the Abstract.

(2) Introduction: The authors have discussed the necessity of studying fire-specific $PM_{2.5}$ and the related health impact in Pacific Asia, and mentioned to use the TFIM method to isolate fire-specific $PM_{2.5}$. It is suggested to briefly summerize previous methods used to estimate fire related air pollution and why the TFIM method is chosen here.

(3) Figure 1: The region has been divided into ESA, NA, CA... in this study. It is recommended for the authors to give the specific scopes of each regions in this figure.

(4) Figures: Please increase resolution of all the figures to enhance clarity, especially enlarging the text in the figures for easier reading by readers.

(5) Line 256-257: The abbreviations "SOS" and "EOS" are unnecessary as they appear only once in the text. The authors should check all abbreviations throughout the manuscript. Full name and abbreviation should be provided upon first mention in the Abstract, main text, table and figures, with abbreviations used in subsequent references.

(6) Figure 4: The figures and its captions should be standalone with enough description to be understood without having to refer to the main text. Authors should provide sufficient information in this figure caption, including what each sub-figure represents.

(7) Line 169: what does "ta positive relationship may exist ..." mean? Please rephrase this sentence.

(8) Line 292-295: Many variations are utilized for estimating fire-specific $PM_{2.5}$ in this study. It would better to list a table showing the specific variations and their information (like resolution and sources) for clarity.

(9) Line 294: The source and details of the GDP data are not provided in Data and Method. Please supplement the information.

(10) Line 305-307: The spatial-temporal resolution of these input features should not completely match the target data. How did the authors correspond them?

---

## Author Comment (AC1)

Dear Editors and Reviewers:

Thank you very much for your careful review and constructive suggestions with regard to our manuscript "The contribution of fires to PM$_{2.5}$ and population exposure in Asia Pacific" (Manuscript Number: egusphere-2025-598). Those comments are valuable and helpful for revising and improving our paper. We have studied these comments carefully and made changes in the manuscript according the reviewers' comments. The responses to the reviewer' comments are listed as follows.

Reviewer #1: This study explores the contributions of fires to air quality in Asia Pacific through isolating fire-specific PM$_{2.5}$ using data-driven method. A comparative analysis of total PM$_{2.5}$ and fire-specific PM$_{2.5}$ variations, along with their effects on public health is conducted. The obvious decline in total PM$_{2.5}$, coupled with a gradual increase in the proportion of fire-specific PM$_{2.5}$, suggests that measures to reduce fires may be a crucial yet under-recognized option for improving air quality and preventing related premature deaths in the future. The authors also reveal differences in spatial distributions of total and fire-specific PM$_{2.5}$. Total PM$_{2.5}$ is mainly concentrated in densely populated areas, while relatively high fire-specific PM$_{2.5}$ disproportionately affect poor populations. These conclusions indicated that the contributions of fires to air quality and health impact are becoming increasingly significant and warrant more attention when developing air pollution standards and control measurements in Asia Pacific. Overall, this study is well-structed, clearly presented, and the results are meaningful. However, some issues still need to be addressed before it can be published:

Response: We would like to thank the reviewer for valuable and affirmative comments of our manuscript. The modifications based on the comments are listed as below.

1. Line 28, Line 34: The full names of abbreviations should be given when their first appearance. What do "VPD" and "PM" stand for?

Response: We feel sorry for the inconvience for reading. "VPD" stands for vapor pressure deficit, while "PM" stands for "particulate matter". We have added the full name of "VPD" and "PM" when their first appearance. Please check Line 17, 31 and 35 in the revised manuscript.

2. Figures: it is recommended to increase the resolution of the figures to enhance clarity and to enlarge the font sizes of annotations and icons within the figures.

Response: Thanks for the suggestion. We have re-plotted all the figures to increase the resolution and enlarge the font size of annotations and icons in the revised manuscript. Please check figures in the revised manuscript and supplementary information.

3. Figure 1: Authors conducted the analysis with several sub-regions in the study, like EA, CA, ESA and NA. However, there are no figures or descriptions indicating the specific areas of each sub-region. It is necessary to add a sub-figure in Figure 1 providing this information.

Response: Thanks for the suggestion. We have renewed Figure 1 by adding Figure 1(b) that indicates the specific areas of each sub region (EA, CA, ESA and NA). Please check Figure 1(b) in the revised manuscript.

4. Line 116-117: Please provide the references regarding the validation of the reliability of MODIS AOD, especially concerning the study regions in this manuscript.

Response: We feel sorry for not including the relevant references to support the viewpoint, and have added five references relating to the validation of the reliability of MODIS AOD concerning the study region or part of the region. Please see Line 136-137, 640-642, 670-673, 682-684 and 718-720 in the revised manuscript.

5. In 2.2 Method: There appears to be a lack of information regarding how the authors align the resolutions of the input and output datasets of machine learning method, considering that the resolutions of different datasets are quite distinct.

Response: We are appreciated with this comment. Although the resolutions of different datasets in the machine learning method are quite distinct, the target data are spatially and temporally dispersed points. Therefore, the construction of machine learning method is essentially point-to-point. The input and output datasets are matched based on their relative positions, meaning that the input data are temporally and spatially closet to the output data. We have added these descriptions regarding the issue of temporal and spatial matching among different datasets. Please check Line 170-174 of the revised manuscript.

6. Figure 4: The line colors in Figure 4 (c) and (d) are somewhat difficult to distinguish, especially since the colors for CA and EA are quite similar, and the two datasets are also close in value, making it hard to interpret. Besides, the figure caption should provide more information about each sub-figures to make the figure stand out of the texts.

Response: We appreciate the comment regarding Figure 4, and have adjusted the line colors in

Figure 4 (as well as in Figure 9, 12 and S1) to ensure that the lines are more distinguishable, especially for CA and EA. Besides, we have added detailed information in figure caption for each sub-figure. Please check Figure 4 and figure caption in the revised manuscript.

7. Line 294: GDP data are included in the machine learning method to estimate the counterfactual $PM_{2.5}$ that is absence of fire influence, however there is no any information describing the data source in the Section 2.

Response: We are sorry for the neglect. The GDP data are obtained from Wang and Sun (2023), measured in PPP 2005 international dollars. We have added the related description about the source of the GDP data and the reference in Line 167-168 and758-759 of the revised manuscript.

8. Figure 7 and Figure 8: The original isolated fire-specific PM2.5 appears to be corresponded to site data shown in Figure 7. However, it is unclear how the authors calculate the fire-specific $PM_{2.5}$ values in each area presented in Figure 8?

Response: We feel sorry for not expressing this issue clearly. The values for each region in Figure 8 are derived from the average values for sites within the region. We have clarified this issue in the caption of Figure 8 and Line 399-400 of the revised manuscript.

9. Line 395-396: "The number of annual premature deaths due to $PM_{2.5}$ ....", does "$PM_{2.5}$" here mean fire-specific $PM_{2.5}$ or total $PM_{2.5}$?

Response: We appreciate the careful comment. The "$PM_{2.5}$" refers to fire-specific $PM_{2.5}$ in the context. We have modified the expression in Line 458 of the revised manuscript accordingly.

10. Line 486-489: Authors should provide some evidence or references supporting the claim of decline in anthropogenic fire alongside increase in wildfires in Asia Pacific. If not, this statement should be omitted to maintain accuracy and credibility.

Response: Thanks for the rigorous comment. We have added relevant references to support the viewpoints discussed, and have rephrased the expressions to eliminate any unverified claims, enhancing the accuracy and credibility of the discussion. Please see Line 549-567 and the references in Line 691-696, 707-709 and 733-734 of the revised manuscript.

Best regards,

Authors

---

## Author Comment (AC2)

Dear Editors and Reviewers:

Thank you very much for your careful review and constructive suggestions with regard to our manuscript "The contribution of fires to $PM_{2.5}$ and population exposure in Asia Pacific" (Manuscript Number: egusphere-2025-598). Those comments are valuable and helpful for revising and improving our paper. We have studied these comments carefully and made changes in the manuscript according the reviewers' comments. The responses to the reviewer' comments are listed as follows.

RC3: This study isolated the fire-specific $PM_{2.5}$ from monitoring concentrations through an observation-driven method, and provides insights into the evolving dynamics of fire specific $PM_{2.5}$ in Pacific Asia. The observed decline in overall $PM_{2.5}$ levels juxtaposed with the rising proportion of fire-specific $PM_{2.5}$ presents a concerning trend, particularly as it suggests the shift in the dominant sources of emissions from anthropogenic activities to more unpredictable fires. This shift not only complicates air quality management efforts but also disproportionately affects vulnerable populations, exacerbating public health risks. Based on the positive correlation between vapor pressure deficit and fire-specific $PM_{2.5}$ highlight the potential exacerbating effects of climate change on future air quality. Overall, this assessment is valuable to call for more attention and researches in the complex interplay of fire-specific air pollution, public health and climate change in the region. However, some issues still need to be improved:

Response: Thank you for the reviewer's insightful comments on our manuscript. We appreciate your recognition of the complexities surrounding fire-specific air pollution, public health, and climate change. Corresponding modifications to address the comments are as following.

(1) Line 28: Please provide the full name of VPD when it appears the first time in the Abstract.

Response: We apologize for the neglect and have added the full name of the abbreviation "VPD" when it first appears in the Abstract. Please see Line 31 of the manuscript.

(2) Introduction: The authors have discussed the necessity of studying fire-specific $PM_{2.5}$ and the related health impact in Pacific Asia, and mentioned to use the TFIM method to isolate fire-specific $PM_{2.5}$. It is suggested to briefly summerize previous methods used to estimate fire related air pollution and why the TFIM method is chosen here.

Response: Thanks for the valuable comment. We have followed the suggestion and added

descriptions in the Introduction, including a comparison of methods for calculating fire-specific PM$_{2.5}$ concentrations and the reason why choosing TFIM in this study. Please see Line 49-64 in the revised manuscript for these updates.

(3) Figure 1: The region has been divided into ESA, NA, CA... in this study. It is recommended for the authors to give the specific scopes of each regions in this figure.

Response: We feel sorry for this neglect and have added Figure 1(b) that gives the specific scopes of each regions. Please check Figure 1(b) in the revised manuscript.

(4) Figures: Please increase resolution of all the figures to enhance clarity, especially enlarging the text in the figures for easier reading by readers.

Response: We apologize for any inconvenience in reading, and have made modifications to enhance clarity and increased the text size within the figures. Please see figures in the revised manuscript and supplementary information.

(5) Line 256-257: The abbreviations "SOS" and "EOS" are unnecessary as they appear only once in the text. The authors should check all abbreviations throughout the manuscript. Full name and abbreviation should be provided upon first mention in the Abstract, main text, table and figures, with abbreviations used in subsequent references.

Response: Thank you very much for the careful comment. We have removed the unnecessary abbreviations from the text, and thoroughly checked all abbreviations used in the manuscript. Please check Line 295-296 in the revised manuscript.

(6) Figure 4: The figures and its captions should be standalone with enough description to be understood without having to refer to the main text. Authors should provide sufficient information in this figure caption, including what each sub-figure represents.

Response: We appreciate your comment and have revised the figure caption to ensure it can standalone and provide sufficient information to be understood without referring to the text. Please see Figure 4 and its caption in the revised manuscript.

(7) Line 169: what does "ta positive relationship may exist ..." mean? Please rephrase this sentence.

Response: We feel sorry for the clerical error and have rephrased the sentence to "the positive relationship may exist ... ". Please see Line 200 in the revised manuscript.

(8) Line 292-295: Many variations are utilized for estimating fire-specific PM$_{2.5}$ in this study. It

would better to list a table showing the specific variations and their information (like resolution and sources) for clarity.

Response: Thanks for the valuable suggestion. We have provided a table that summarizes the original input features for estimating fire-specific $PM_{2.5}$, including details on their resolutions and sources to facilitate a comprehensive overview of data utilized. Please see Table 1 in the revised manuscript.

(9) Line 294: The source and details of the GDP data are not provided in Data and Method. Please supplement the information.

Response: We apologize for the neglect and have provided the source of GDP data in Line 167-168 of the revised manuscript.

(10) Line 305-307: The spatial-temporal resolution of these input features should not completely match the target data. How did the authors correspond them?

Response: We are thankful for the comment. The target data consist of temporally and spatially inconsistent points. Therefore, we match the multi-source input data with the target data based on their distances and construct the machine learning model point-to-point. Detailed descriptions can be found in Line 170-174 of the revised manuscript.

Best regards,

Authors

---

## Author Comment (AC3)

Dear Editors and Reviewers:

Thank you very much for your careful review and constructive suggestions with regard to our manuscript "The contribution of fires to PM$_{2.5}$ and population exposure in Asia Pacific" (Manuscript Number: egusphere-2025-598). Those comments are valuable and helpful for revising and improving our paper. We have studied these comments carefully and made changes in the manuscript according the reviewers' comments. The responses to the reviewer' comments are listed as follows.

RC2: This study investigated the contribution of forest and vegetation fires to PM$_{2.5}$ and public health across Pacific Asia. The fire-specific PM$_{2.5}$ was effectively isolated from the monitoring data using the trajectory-fire interception method (TFIM) in combination with the random forest method. Based on the reliable and timely dataset, the geographical disparities in population exposure to both PM$_{2.5}$ and fire-specific PM$_{2.5}$ were estimated. The topic is novel, and the findings provide valuable insights for PM$_{2.5}$-related public health. Moreover, this manuscript is well-structured and readable. However, several issues need to be clarified before it can be considered for publication. Below are specific comments:

Response: Thanks for recognition of novelty and value of our study. We appreciate the valuable comments and suggestions and have made necessary revisions of our manuscript as below.

Specific Comments

1. I suggest changing the regional abbreviations to Southeast Asia (SEA), Northeast Asia (NEA), East Asia (EA) and Central Asia (CA), respectively. And these key regions should be marked in Figure 1.

Response: Thank you for the suggestion. We have updated the regional abbreviations to Southeast Asia (SEA), Northeast Asia (NEA), East Asia (EA), and Central Asia (CA) to enhance clarity. Besides, we have added Figure 1(b), which provides specific scopes of each region. Please see Figure 1(b) and the corresponding text thorough the revised manuscript for the modifications.

2. To derive fire-specific PM$_{2.5}$ concentrations, a machine learning method was employed using aerosol variables, AOD, meteorological factors, land use, NDVI, and the GDP data. Ultimately, 15 of these variables were selected to fit PM$_{2.5}$ concentrations. In recent years, many Asian countries have implemented strict anthropogenic emissions reduction strategies to mitigate PM$_{2.5}$ pollution

(see lines 322-330 for details). Anthropogenic emissions play crucial role in $PM_{2.5}$, but they are not incorporated into the machine learning model. While some variables, such as GDP and population, can indirectly reflect changes in anthropogenic emissions, the relationship needs to be further clarified. Therefore, the authors should briefly describe the machine learning method and variables used, even in the supporting information.

Response: We appreciate the insight comment. It is indeed important to acknowledge the significant role of anthropogenic emissions in ambient $PM_{2.5}$ concentrations across Asian countries. To comprehensively account for anthropogenic aerosols in this study, we considered not only indirect reflection features, such as GDP and population, during the construction of machine learning model, but also various aerosol data that directly reflect anthropogenic sources. This includes black carbon, organic carbon, $SO_2$ surface mass concentrations and so on. These data are derived from the MERRA-2 reanalysis, which assimilates multiple aerosol remote sensing, emissions, and meteorological datasets using the Goddard Earth Observing System Model. With these advances, MERRA-2 aerosol products can provide reliable anthropogenic and natural aerosols (like dust). Therefore, we did not incorporate anthropogenic emissions during dataset construction. We have added related descriptions about the machine learning method and variable used. Please see Line 234-242 and 347-356 in the revised manuscript.

3. Section 3 Results: It is recommended that additional subheadings (e.g., 3.1, 3.2, 3.3, ...) be added to improve clarity. Furthermore, more attention should be given to key regions (SEA, EA, NEA and CA) during wildfires. Regions with extremely sparse stations (Figure 1), such as Mongolia and the Tibetan Plateau, may be masked out due to the large uncertainties in their results.

Response: Thank you very much for the valuable comment and suggestions.

(1) We have added subheadings in Section 3 Results to enhance clarity and facilitate reading. The subheadings are:

3.1 Estimating fire-specific $PM_{2.5}$

3.2 The spatial and temporal distributions of $PM_{2.5}$ and fire-specific $PM_{2.5}$

3.3 The fire-specific $PM_{2.5}$ exposure and health impact

3.4 Future trends of fire-specific $PM_{2.5}$ under climate change

The subheadings help structure the section more clearly and improve the overall readability.

(2) Regarding the emphasis on key regions (SEA, EA ,NEA and CA) during fires, we have provided Figure 8, 9 and S1, along with analysis in the text, revealing spatial and temporal distributions of fire-specific $PM_{2.5}$ in the key regions during fire season. Additionally, we have showed and discussed the population exposure and health impact of fire-specific $PM_{2.5}$ in key regions during fire season.

(3) As for the regions with extremely sparse stations, like Mongolia and Tibetan Plateau. We think the sparsity of stations will not increase the uncertainty in the calculations for the specific stations. There are two key steps in estimation of fire-specific $PM_{2.5}$ at a station using TFIM method.

First, determine whether a station was exposed to fire smoke at a specific time based on back trajectory calculations and satellite fire point data. The fire point distribution monitoring by satellite is not influenced by distribution of ground monitoring stations, and reflects exposure conditions at the station accurately. Subsequently, extract non-smoke $PM_{2.5}$ employing machine learning method. In this process, we correspond data of different resolutions to station data based on the nearest distance principle. The model was constructed through point-to-point comparisons.Both these key processes remain unaffected by the distribution of stations.

In areas with sparse stations, while the calculation results may not accurately reflect the fine spatial distribution within the region, using these averages to represent the regional mean is still relatively reasonable.

We have added some explanations regarding calculation and representative of fire-specific $PM_{2.5}$ in regions with sparse stations. Please see subheadings in Result,    Figure 8, 9 and S1 and Line 400-403 of the revised manuscript.

4. Figure 10 and 11: To my knowledge, the Health Impact Function (HIF) exhibits large uncertainty due to the relative risk (RR). Additionally, the Integrated Exposure-Response (IER) equation varies by region and population, resulting in a confidence interval in the estimated number of premature deaths. Although these key parameters are from Burrett et al. (2014) and Song et al. (2017) (please note that the citation of Song is missing in the reference list) in this study, they should still be explicitly provided.

Response: We appreciate the insightful comment and apologize for the missing information in the reference list. We have included the confidence intervals in estimated number of premature deaths,

as the key parameters from Burrett et al. (2014) and Song et al. (2017) provide necessary confidence intervals for these values. Additionally, the parameters used in this study have been updated in the Supplemental Information, and the missing citation for Song et al. (2017) has been added to the reference list. Please see Line 23-28, 264-265, 440-447, 528-533 and 752-753 in the revised manuscript, and Table S1 in the Supplemental Information.

5. Figure 12: The results related to vaper pressure deficit (VPD) appear some what disconnected from the main body of this manuscript.

Response: Thanks for the comment. We provide variations of VPD in Figure 12 is due to its common use as a climate indicator to establish the relationship between climate change and fire occurrence or emissions (Abatzoglou and Williams, 2016; Burke et al., 2023). In this study, we analyzed historical data and found the positive relationship existing between VPD and fire-specific $PM_{2.5}$ across different regions of Asia Pacific (Figure 12a). Based on this, we can roughly infer future trend in fire-specific $PM_{2.5}$ through examining the VPD future trends (Figure 12b), assuming that relationship between future VPD and fire-specific $PM_{2.5}$ continues to exist. The analysis is important because because they allow us to associate fire-specific $PM_{2.5}$ with climate change, and explore future changes in fire-specific $PM_{2.5}$. Through discussion, we can examine the necessity of addressing climate change and air pollution in a coordinated manner. Of course, studying the future trends of fire-specific $PM_{2.5}$ will require integrating more data and methods for a more precise analysis, which is a direction for our future research. We have added more descriptions about this in the manuscript. Please see Line 557-563 in the revised manuscript.

Technical Corrections

1. Normally, references are sorted alphabetically rather than chronologically.

Response: Thank you for the careful comment. We have re-sorted the references to ensure they are sorted as alphabetically. Please see Reference in the revised manuscript.

2. Line 230: "in this study" instead of "is this study"

Response: We apologize for the clerical error and have modified it in Line 270 of the revised manuscript.

3. Figure 8: Please add the unit.

Response: We feel sorry for the neglect and have added the unit in the figure. Please see Figure 8 in the revised manuscript.

4. The manuscript contains minor spelling and grammatical errors that should be corrected.

Response: Thanks for the valuable comment. We have carefully review the text and make corrections to ensure clarity and accuracy. Please see throughout the revised manuscript.

Best regards,

Authors

---

## Referee Report (RR1)

**Review of "The contribution of fires to PM$_{2.5}$ and population exposure in Asia Pacific" by Lu et al.**

**General Comments**

I appreciate the work that the authors have put into revising the manuscript and responding to my previous comments. The authors have thoroughly addressed my concerns. The manuscript is well-structured and complete, and I believe it is suitable for publication in Atmospheric Chemistry and Physic once it is approved by other reviewers.